# Interleukin-6 Gene Expression Changes after a 4-Week Intake of a Multispecies Probiotic in Major Depressive Disorder—Preliminary Results of the PROVIT Study

**DOI:** 10.3390/nu12092575

**Published:** 2020-08-26

**Authors:** Alexandra Reiter, Susanne A. Bengesser, Anne-Christin Hauschild, Anna-Maria Birkl-Töglhofer, Frederike T. Fellendorf, Martina Platzer, Tanja Färber, Matthias Seidl, Lilli-Marie Mendel, Renate Unterweger, Melanie Lenger, Sabrina Mörkl, Nina Dalkner, Armin Birner, Robert Queissner, Carlo Hamm, Alexander Maget, Rene Pilz, Alexandra Kohlhammer-Dohr, Jolana Wagner-Skacel, Kathrin Kreuzer, Helmut Schöggl, Daniela Amberger-Otti, Theresa Lahousen, Birgitta Leitner-Afschar, Johannes Haybäck, Hans-Peter Kapfhammer, Eva Reininghaus

**Affiliations:** 1Department of Psychiatry and Psychotherapeutic Medicine, Medical University of Graz, Auenbruggerplatz 31, 8036 Graz, Austria; alexandra.rieger@medunigraz.at (A.R.); frederike.fellendorf@medunigraz.at (F.T.F.); martina.platzer@medunigraz.at (M.P.); mat.seidl@icloud.com (M.S.); lillimendel@posteo.de (L.-M.M.); renate.unterweger@klinikum-graz.at (R.U.); melanie.lenger@medunigraz.at (M.L.); sabrina.moerkl@medunigraz.at (S.M.); nina.dalkner@medunigraz.at (N.D.); armin.birner@medunigraz.at (A.B.); robert.queissner@medunigraz.at (R.Q.); carlo.hamm@medunigraz.at (C.H.); alexander.maget@stud.medunigraz.at (A.M.); rene.pilz@medunigraz.at (R.P.); alexandra.kohlhammer-dohr@klinikum-graz.at (A.K.-D.); jolana.wagner-skacel@medunigraz.at (J.W.-S.); kathrin.kreuzer@medunigraz.at (K.K.); helmut.schoeggl@klinikum-graz.at (H.S.); daniela.amberger-otti@klinikum-graz.at (D.A.-O.); theresa.lahousen@medunigraz.at (T.L.); birgitta.leitner-afschar@klinikum-graz.at (B.L.-A.); hanspeter.kapfhammer@medunigraz.at (H.-P.K.); eva.reininghaus@medunigraz.at (E.R.); 2Department of Mathematics & Computer Science, University of Marburg, 35043 Marburg, Germany; hauschild@uni-marburg.de; 3Institute for Pathology, Neuropathology and Molecular Pathology, Medical University of Innsbruck, 6020 Innsbruck, Austria; anna.birkl-toeglhofer@i-med.ac.at (A.-M.B.-T.); johannes.haybaeck@i-med.ac.at (J.H.); 4Institute of Psychology, University of Bamberg, 96047 Bamberg, Germany; tanja.faerber@uni-bamberg.de

**Keywords:** depression, gut-brain-axis, probiotics, gene expression, interleukin-6

## Abstract

Major depressive disorder (MDD) is a prevalent disease, in which one third of sufferers do not respond to antidepressants. Probiotics have the potential to be well-tolerated and cost-efficient treatment options. However, the molecular pathways of their effects are not fully elucidated yet. Based on previous literature, we assume that probiotics can positively influence inflammatory mechanisms. We aimed at analyzing the effects of probiotics on gene expression of inflammation genes as part of the randomized, placebo-controlled, multispecies probiotics PROVIT study in Graz, Austria. Fasting blood of 61 inpatients with MDD was collected before and after four weeks of probiotic intake or placebo. We analyzed the effects on gene expression of tumor necrosis factor (*TNF*), nuclear factor kappa B subunit 1 (*NFKB1*) and interleukin-6 (*IL-6*). In *IL-6* we found no significant main effects for group (*F*_(1,44)_ = 1.33, *p* = ns) nor time (*F*_(1,44)_ = 0.00, *p* = ns), but interaction was significant (*F*_(1,44)_ = 5.67, *p* < 0.05). The intervention group showed decreasing *IL-6* gene expression levels while the placebo group showed increasing gene expression levels of *IL-6*. Probiotics could be a useful additional treatment in MDD, due to their anti-inflammatory effects. Results of the current study are promising, but further studies are required to investigate the beneficial effects of probiotic interventions in depressed individuals.

## 1. Introduction

Major depressive disorder (MDD) is a common psychiatric disorder with a lifetime prevalence of approximately 16–20%. Symptoms include depressed mood, lack of energy, anhedonia, sleep disorders, changes in appetite, and cognitive deficits [1]. More than 33 million people in the European Union and over 300 million people worldwide are affected by MDD according to the World Health Organization (WHO) [2] but only one third respond directly to the anti-depressive medication [3]. MDD is a potentially life-threatening disease and suicide is even the second leading cause of death in 15−29-year-olds. The estimated economic burden of MDD in the European Union was €136.3 billion in 2007 due to reduced productivity and health service uptake [4]. Thus, new treatment options are crucial to help treating those affected by MDD.

According to previous literature MDD is caused by a network of biopsychosocial factors with a high genetic burden. Concordance rates for monozygotic twins lie at approximately 50% and 15-20% for dizygotic twins [5]. Hence, the polygenic, complex genetic predisposition is important for disease mechanisms, multiple gene-environment interactions are necessary for disease onset. Diverse molecular biological mechanisms are involved in the pathogenesis of MDD e.g. epigenetics-, gene-expression-, metabolomics- and gut-brain-axis mechanisms, which are summarized in diverse comprehensive reviews [6,7,8,9]. Previous studies constantly showed an association between depression and chronic inflammation, as well as oxidative stress [10,11,12,13]. Oxidative stress and chronic inflammation are commonly associated with obesity, which is more frequent in mood disorders such as MDD or BD [14,15]. This implies a crucial role for inflammatory pathways in MDD and refers to the so-called “inflammation model” of MDD.

During the process of an inflammatory response, cytokines act as messengers between immune cells, both in the pro-inflammatory and the anti-inflammatory way. Besides the prominent role of interleukin-6 (IL-6) in infection, cancer and inflammation [16,17], IL-6 appears to be a cytokine, which mediates an interaction between the immune system and CNS (central nervous system) function [18]. For instance, serum levels of the soluble IL-6 receptor have been shown to increase at the onset of sleep. Additionally, IL-6 is supposed to have hormone-like attributes and might influence the neuroendocrine system as well as neuropsychological behavior [19]. Stress response, prostaglandins and adipokines were shown to serve as activators of IL-6 expression [19]. Increased levels of IL-6 in serum and/or plasma concentration, acute-phase-proteins, expression of chemokines and adhesion molecules have been commonly observed in MDD patients [20]. Incident depression was significantly associated with increases in IL-1β, IL-6, and IL-8 serum levels in late-life depression [21]. Altered pro-inflammatory cytokine levels are present in other psychiatric diseases, too. A meta-analysis of blood cytokine alteration described the same pattern of elevated IL-6 levels in schizophrenia and bipolar disorder [22]. In addition, the post-traumatic stress disorder (PTSD) is associated with increased levels of IL-6, interleukin-1 beta (IL-1β), interferon gamma (INFγ) and tumor necrosis factor alpha (TNFα) [23]. TNF signaling plays a major role in MDD according to a recent Weighted Gene Coexpression Network Analysis [24]. The soluble TNF-α receptor type II could also play a role as biomarker for discriminating MDD and healthy controls and was suggested in a panel of 9 biomarkers in a recent Consensus paper of the WFSBP (World Federation of Societies of Biological Psychiatry) Task Force on Genetics [25]. 

Thus, previous literature shows that MDD mechanisms include mild inflammatory processes, which can be triggered by diverse gene-environment interactions. For example, inflammatory processes can be generated by the so-called “leaky-gut-syndrome”, a mechanism of the gut-brain axis, which results from increased gut permeability [26]. Zonulin is a marker of the leaky-gut-syndrome [27,28] and was associated with affective disorders, which shows the strong connection between mood, inflammation, and the gut-brain axis [29]. Importantly, high IL-6 levels have also been shown in study participants having high zonulin [30]. The presence of dysbiosis, causing the breakdown of the intestinal permeability, can lead to an inflammatory condition which is not limited to the gut, since the pro-inflammatory cytokines can get into the bloodstream and reach the brain [31]. Therefore, mechanisms of the gut-brain axis can affect systemic inflammation and it is not surprising that the gut-brain-axis has been associated with MDD in multiple studies [32,33,34,35]. The term “microbiota” includes all bacteria, viruses, fungi, and other single-celled organisms living in a certain environment such as skin, mouth, lungs or gut. “Microbiome” refers to all their genetic material [36]. To date the Flemish Gut Flora Project was one of the largest available studies in MDD microbiome research [37]. They found a reduction of Coprococcus spp. and Dialister levels in patients with MDD and discovered a positive correlation between “quality of life”, Faecalibacterium and Coprococcus levels, and the “gut microbiomes’ ability to synthesize certain dopamine metabolites” [37]. Depressive phenotypes could also be transferred from one rat to another by transferring the microbiome [34].

Probiotics that are supposed to change the diversity of the microbiome and their metabolites, could potentially affect inflammation markers in the blood and consequently depression, because the gut-brain axis can affect pro-inflammatory cytokine levels in the blood stream as described before [31]. According to the Food and Agriculture Organization (FAO) and the WHO, probiotics are live strains of microorganisms that benefit the hosts health when administrated in an adequate amount [38]. Probiotics exert their beneficial effects through various mechanisms of action: (1) interference with potential pathogens, (2) improvement of barrier function, (3) immunomodulation and influence on other organs of the body through the immune system and (4) neurotransmitter production [39,40]. A variety of studies has shown that probiotics can also affect IL-6 levels and general inflammation pathways [41,42,43]. A recent meta-analysis by Milajerdi et al. analyzed the effects of probiotics on inflammatory biomarkers in serum [44]. They included a total of 42 placebo-controlled studies with healthy subjects and patients suffering from different diseases. For IL-6 a significant reduction in serum levels was found after probiotic supplementation, based on 16 studies. Furthermore, effect sizes from 18 studies revealed a significant reduction in serum TNF concentrations after probiotic supplementation [44]. 

Nevertheless, sufficiently large studies investigating the effects of probiotics in psychiatric disorders with randomized controlled trials (RCTs) are lacking. So far, only a few RCTs investigating the effects of probiotics and micronutrients are available. A placebo-controlled, randomized study of 40 young, mentally healthy adults was able to show that special multi-species probiotics were able to improve negative cognitive reactions to stressful situations [45]. Furthermore, it could be demonstrated that the administration of probiotics in healthy people led to reduced depressive and anxiety symptoms [46]. There is only a hand full of RCTs, which investigated the effects of probiotics and micronutrients in psychiatric disorders. For example, Akkasheh and colleagues examined 40 depressed patients [47], Rucklidge et al. investigated the safety of long-term consumption of micronutrients for the treatment of psychiatric symptoms [48], Tomasik et al. analyzed probiotics in a sample with individuals with schizophrenia [49], and our research group investigated beneficial effects of probiotics in euthymic individuals with bipolar disorder [50,51]. Results showed decreased depression scores after two months of probiotic intake with three different species (Lactobacillus acidophilus, Lactobacillus casei, Bifidobacterium bifidum) [47], or significant improvement in performance of attention, psycho-motor speed and executive function after three months of probiotic supplementation [50].

In the present PROVIT study, we analyzed the effects of multispecies probiotics on mood, cognitive function, gut microbiome and gene expression in patients with MDD [52]. The current original report shows preliminary results from our PROVIT study, including gene expression data, measured with quantitative polymerase chain reaction (qPCR) of inflammation-related genes, namely IL-6, TNF, and NFKB1. We hypothesized that gene expression of inflammatory genes (IL-6, TNF, and NFKB1) would decrease in the intervention group taking the multispecies probiotic drink (‘OMNi-BiOTiC® STRESS Repair’) over four weeks.

## 2. Methods 

### 2.1. Study Design

The PROVIT study is a monocentric, randomized, placebo-controlled trial that aims to investigate the effects of multispecies probiotics on depressive symptomatology, cognitive function, microbiome, blood parameter, and gene expression. The patients were recruited at the Department of Psychiatry and Psychotherapeutic Medicine at the Medical University Graz (MUG). Inpatients with MDD, who were admitted for treatment as usual to our ward, received over 4 weeks a multispecies-probiotics- or placebo drink, which was provided in the morning before breakfast by the doctors on call. The study was accepted by the local ethics board (EK 29-235 ex 16/17) and is registered at clinicaltrials.com (NCT03300440). Inclusion criteria for PROVIT included written consent after previous written and verbal information, diagnosis of MDD by the treating psychiatrist and age between 18 and 75 years. Exclusion criteria included acute suicidality, lack of consent, pregnancy or breastfeeding, severe active drug dependence (i.e., alcohol, benzodiazepines, morphine), other currently active severe mental/cerebral organic disease (e.g., epilepsy, brain tumor), severe skull-brain trauma/brain surgery in the past, known florid tumor disease, congenital/infantile mental disability, moderate/severe dementia, severe florid autoimmune diseases or current immunosuppression (e.g., lupus erythematosus, human immunodeficiency virus (HIV), multiple sclerosis), antibiotic therapy within last month, chronic laxative abuse, acute infectious diarrheal disease, regular intake of butyrate-containing or probiotic supplements in the last year, no intake of (other) probiotics and prebiotics during the entire trial, no intake of antibiotics or prebiotics during the entire trial. Patients were not excluded based on the consumption of dairy products before the PROVIT study in the past. Computerized block randomization was used to assign the individuals to either intervention or placebo groups. In total, 82 patients were randomized and allocated to a group (see Figure 1, [53]). Of these, 61 individuals completed the study. Reasons for dropout were e.g., early discharge, non-compliance, or use of antibiotics during the study period.

### 2.2. Demographics and Scales of PROVIT

The diagnostic structured interview M.I.N.I interview (Mini-International Neuropsychiatric Interview, [54]) was performed during admission by a psychiatrist to confirm the diagnosis of MDD. The PROVIT study included among other the Hamilton depression scale (HAMD, [55]) and the Beck Depression Inventory-II (BDI-II, [56]). The HAMD is an external assessment to evaluate the severity of depressive symptoms, including 21 items on a three-point or five-point Likert-type scale [55]. BDI-II is a self-report inventory to assess symptoms of acute depression in the last two weeks. It also contains 21 items, each answer being scored on a scale value of 0 to 3 points [56]. Moreover, clinical and demographic parameters including age, weight, height, sex and medication as well as a cognitive test battery and lifestyle questionnaires were assessed at day one and day 28. Furthermore, stool and fasting blood samples were collected at both time points. The workflow and design of the PROVIT study are depicted in Figure 2.

### 2.3. Probiotic Intervention

Individuals in the intervention group received the multispecies-probiotic ‘OMNi-BiOTiC^®^ STRESS Repair’ (by Institute AllergoSan). Individuals in the placebo group received biotin/vitamin B 7 as a drink with the same color, consistency and taste as the probiotic drink. OMNi-BiOTiC^®^ STRESS Repair’ includes nine bacterial strains with at least ≥2.5 × 10^9^ colony forming units (CFU) per gram. One drink contained 3 g, which gives a total number of at least 7.5 × 10^9^ CFU per bag. The bacterial strains are: *Lactobacillus casei W56, Lactobacillus acidophilus W22, Lactobacillus paracasei W20, Bifidobacterium lactis W51, Lactobacillus salivarius W24, Lactococcus lactis W19, Bifidobacterium lactis W52, Lactobacillus plantarum W62, Bifidobacterium bifidum W23*. Moreover, the product consists of maize starch, maltodextrin, inulin, potassium chloride, magnesium sulfate, fructooligosaccharides (FOS), enzymes (amylases), manganese sulfate and bacterial strains. Both groups received biotin, because all patients should receive a substance that might be beneficial for them. The study substances (3 g, 7.5 × 10^9^ CFU) were stirred with water and then left to rest for an activation time of 10 minutes. All participants received the mixed drinks every morning at 7 am before breakfast from the on-call medical doctor. During the intervention phase, patients did not take other prebiotics, antibiotics or laxatives. Like any other inpatient at the Department of Psychiatry and Psychotherapeutic Medicine, participants received treatment as usual, which included physiotherapy, occupational therapy, psychopharmacological therapy and psychotherapy. If required pharmaceuticals were changed or adapted. The category of psychopharmacological medication, the quantity and changes were logged.

### 2.4. Gene Expression Methods

Fasting blood was taken with PAXgene blood RNA tubes (Nr 762165, PreAnalytix GmbH, Hombrechtikon, Switzerland) for RNA isolation from peripheral blood mononuclear cells (PBMCs). After taking blood, PAXgene blood RNA tubes were tilted 10 times, incubated at room temperature for one hour according to the manufacturer’s SOPs and frozen at −80 degrees Celsius. 

Total RNA was isolated from at −80 degrees Celsius frozen blood stored in PAXgene Blood RNA tubes using the PAXgene Blood RNA Kit (PreAnalytix GmbH, Hombrechtikon, Switzerland) according to the manufacturer’s instructions. The RNA quantity and quality were measured on the NanoDrop 1000 Spectrophotometer (Thermo Fisher Scientific, Waltham, MA, USA). cDNA was reversely transcribed using the High-Capacity cDNA Reverse Transcription Kit (Applied Biosystems, Foster City, USA) containing RNAse inhibitor (Applied Biosystems, Foster City, CA, USA) according to the manufacturer’s instructions. Quantitative real-time polymerase chain reaction (PCR) was performed on the QuantStudioTM 7 Flex Real-Time PCR System (Applied Biosystems, Foster City, CA, USA), using the Luna Universal qPCR Master Mix (New England Biolabs, Ipswich, MA, USA), 200 nM of the primers or when applicable the QuantiTect Primer Assay (Qiagen, Hilden, Germany) and 2.5 ng of the cDNA. The following primers were used for the inflammation genes and the endogenous control genes: *IL-6* forward 5′-GGCACTGGCAGAAAACAACC-3′ and reverse 5′-GCAAGTCTCCTCATTGAATCC-3′; *TNF* forward 5′-CGAACCCCGAGTGACAAG-3′ and reverse 5′-CTGGTAGGAGACGGCGATG-3′; *NFKB1* forward 5′-AACAGAGAGGATTTCGTTTCCG-3′ and reverse 5′-TTTGACCTGAGGGTAAGACTTCT-3′; QuantiTect Primer Assay (QT00000721, Qiagen, Hilden, Germany). The cycling conditions were as follows: 50 °C for 2 min and 95 °C for 10 min for the initial denaturation, followed by 45 cycles of denaturation at 95 °C for 15 s and extension at 60 °C for 60 s. The sample’s melting curves were measured to confirm the specificity of amplified products. All samples were processed as technical triplicates. The gene expression was determined by the 2−ΔΔCt method [57]. The mean Ct value of GAPDH and TBP was used for normalization of transcript abundance. The expression values represent the x-fold change.

### 2.5. Statistics

Differences between the two groups at baseline were analyzed with the *t*-test, Mann–Whitney-U test or chi-square tests depending on normal distribution, Kolmogorov–Smirnov test, and scale level. Therefore, sex, smoking habit, and the use of dairy products were analyzed with the chi-square test. Differences in age and waist-to-hip ratio were analyzed with *t*-tests. Years of education, illness duration and BMI were not normally distributed and therefore analyzed with Mann–Whitney-U tests. We calculated two repeated measures two-way-analysis of variances (ANOVA) with the independent factors time point (t1 vs. t2) and group (intervention vs. placebo). The dependent variables are scores in BDI-II and HAMD respectively levels of gene expression in *IL-6, TNF*, and *NFKB1*. Statistics were analyzed with SPSS 25.0 (Armonk, NY, USA: IBM Corp, 2017), the level of significance was set to *p* < 0.05.

### 2.6. Network Analysis

We used a set of established methods from systems biology, proven to be highly valuable and successful in identifying disease-related protein relationships and functional pathways, particularly in other areas of research (i.e., oncogenomics) [58]. At First, we investigated the protein-protein interactions (PPIs) of the genes of interest (IL-6, TNF and NFKB1), by querying the Integrated Interaction Database (IID) [59]. This database allows extracting tissue informed PPI networks, i.e. by focusing on interactions observed in brain tissues, enabling a fine-grained evaluation of which biological processes are involved. Subsequently, to visualize and present these discovered brain-tissue PPI networks we employed the Network Analysis and Visualization, and Graphing Toronto (NAViGaTOR) [60,61]. NAViGaTOR supports automated integration of multiple databases (i.e. interaction and context annotation) and includes a large number of evaluation, analysis, and visualization approaches. Moreover, the resulting interacting proteome is the starting point for a more finegrained pathway enrichment analysis using g:Profiler [62,63]. Finally, the resulting annotation and contextual information can be visualized and interpreted using Cytoscape [64] and the Enrichment Map app [65]. This analysis highlights underlying molecular relationships of nutrition and MDD and potentially generate novel targets of interest in this area.

## 3. Results

### 3.1. Descriptive Statistics

In total 28 individuals (20 women) from the intervention group and 33 individuals (27 women) from the placebo group completed the study. At baseline, there were no significant differences between the two groups in age, sex, body mass index (BMI), waist-to-hip ratio (WHR), years of education, or illness duration. The only difference between the groups at baseline was found for smoking. There were significantly more smokers in the placebo group (see Table 1).

Detailed psychopharmacological medication is presented in Table 2. There was no significant difference in the number of substance classes (*χ^2^*(2) = 3.169, *p* =0.205) between the two groups. In the intervention group six, whereas in the placebo group two individuals were not pre-medicated. Fourteen patients in the intervention group respectively nineteen participants in the placebo group took medication from one or two different substance categories and eight respectively twelve individuals took drugs from three or more distinct substances.

The results of the repeated measure ANOVA with the two independent factors time point (beginning and end of the study) and group (intervention vs. placebo) and dependent variables BDI-II and HAMD showed that both groups significantly improved in psychological parameters over time (*F*_(2,55)_ = 60.79, *p* < 0.001; BDI: *F*_(1,56)_ = 114.64, *p* < 0.001; HAMD: *F*_(1,56)_ = 47.85, *p* < 0.001; see Table 3). There was no significant effect for group (*F*_(2,55)_ = 2.69, *p* = ns), nor a significant time group interaction (*F*_(2,55)_ = 0.17, *p* = ns). 

### 3.2. Gene Expression

We analyzed gene expression levels again with a repeated measure ANOVA with the between subject factor group and the within subject factor time point. There were no significant main effects for time (*F*_(3,42)_ = 0.367, *p* = ns), nor group (*F*_(3,42)_ = 0.076, *p* = ns). On a univariate level no significant effects were found in *TNF*, nor *NFKB1*. We neither found significant main effects for group nor time in *IL-6* gene expression levels, but the interaction effect was significant (see Table 4). Patients in the probiotics group decreased, while patients in the placebo group increased in *IL-6* gene expression levels (see Figure 3). Results did not change when adjusted for smoking. 

### 3.3. Network Analysis Results

The query of the integrated interaction database for proteins that are interacting with the three target genes *IL-6*, *NFKB1* and *TNF* in brain tissue resulted in a brain-PPI network containing 458 interacting genes (see Figure 4). The visualization and annotation of this PPI network (using NAViGaTOR) shows that interacting proteins are involved in biological processes such as building up cellular components, organization of biogenesis, developmental processes and growth, immune system processes, locomotion, metabolic processes, rhythmic processes, signaling and single-organism processes. 

A more detailed description of the function of the protein-protein network of our analyzed target genes is depicted in the enrichment map (see Figure 5 and enrichment map in the Appendix A). The GO-biological processes in the enrichment map give a more specific idea of the enriched processes of *IL-6*, *TNF*, *NFKB1*. The potential differential expression of the target genes by probiotics could therefore affect the whole interaction network, which includes interesting potential mechanisms that could be affected by probiotics (see Figure 5, excerpt of the enrichment map). Cellular response to stress, immune system response, cellular response to bacteria, regulation of response to external stimuli and response to cytokines, as well as response to chemicals could be potentially interesting in probiotics research and should be further examined in future investigations.

## 4. Discussion

In this monocentric, randomized, placebo-controlled, multispecies probiotics study we aimed to investigate the effects of a multispecies-probiotic supplementation on gene expression of inflammatory genes in depressive patients. The findings clearly indicate that patients with a four-week probiotic intake showed decreasing IL-6 gene expression levels while the placebo group showed increasing gene expression levels of IL-6. This result confirms the positive effects of multispecies probiotics on mild inflammation in depressive disorders. 

IL-6 is thought to play an important role in the pathogenesis of depression and previously, elevated levels were associated with bad prognosis and worse disease course [66]. Data from the longitudinal Netherlands Study of Depression and Anxiety suggests cross-sectional and bidirectional longitudinal associations between depression and IL-6 levels in blood plasma. Depression at baseline was associated with higher IL-6 levels at baseline and with higher IL-6 levels after the two- and six-year follow-ups. Furthermore, higher levels of baseline IL-6 predicted depression chronicity over time [67]. Previous studies investigating other clinical samples showed that probiotics could affect general inflammation pathways. Borzabadi et al. demonstrated that a 12-week probiotic supplementation in patients with Parkinson’s disease significantly improved gene expression of IL-1, IL-8, TNF-α, TGF-β and PPAR-γ [41]. In patients with multiple sclerosis a 12-week trial with probiotics led to a significantly improved gene expression of IL-8 and TNF-α [43]. A recent meta-analysis among patients with diabetes showed that probiotic and synbiotic supplementation significantly decreased TNF-α and C-reactive protein, but there were no effects on IL-6 serum levels [68]. 

Compared to non-depressed controls, patients with MDD show specific alterations of several taxa at family and genus levels, specifically the family of Prevotellaceae, genus Coprococcus and Faecalibacterium, which were shown to be decreased [69]. Due to this alterations, probiotics were thought to exert positive effects on the gut microflora and continue to be studied in randomized controlled trials. Indeed, recent studies showed that special multi-species probiotics were able to improve negative cognitive reactions to stressful situations [45] and that the administration of probiotics in healthy people led to a reduction in depressive and anxiety symptoms [46]. Akkasheh and colleagues examined 40 depressed patients who received probiotics for two months in a randomized study. There was a significant improvement in depression scores, in addition there were significant improvements in insulin and CRP values and in oxidative stress parameters in the active treatment group compared to the placebo group [47]. Unfortunately, we could not show positive intervention effects on symptoms of depression and psychological distress. This is opposing the results of meta-analysis concluding that while probiotics improved depressive symptoms in patients with severe depression, this improvement could not be seen in non-depressed individuals. Several reasons could be responsible for this finding: the duration of the probiotic application or the applied dosage could have been to low as some antidepressants are known to have anti-microbial effects [70]. Another possible explanation could be that probiotics only work for special subtypes of depression such as the inflammatory subtype. It is conceivable that the beneficial effects of probiotics may only be effective on psychological parameters in less severe states of MDD. However, a meta-analysis by Ng et al. (2018) with 10 clinical trials showed that only individuals with mild to moderate depression improved symptoms, while there were no significant effects in healthy individuals [71]. A recent meta-analysis of randomized controlled trials indicates a significant improvement of depressive symptomatology followed by probiotic supplements [69]. Although there are some promising results in depression, further randomized controlled trials are needed to determine their efficacy.

There are some limitations to our study. First, we could not take into account effects of different medication. All patients received various antidepressants and the medication was adjusted during their hospitalization. We recorded all drugs, but it was not possible to statistically control for that (see Figure 2 for the medication of the present sample). Second, the period of the probiotic supplementation of four weeks could be too short to show effects. Third, we used a multispecies-probiotic product with nine bacterial strains. Studies have shown that multispecies probiotics are generally considered to have higher efficacy. For example, in Lactobacillus only trials no significant differences of depressive symptoms were shown and there was a significant difference in effect size between lactobacillus only trials and others [72]. However, as we know, the gut microbiome is not the same in all individuals, the probiotic supplements ought to be more personalized. Further studies are required to investigate the beneficial effects of probiotic interventions regarding duration, dosage, and bacterial strains. Fourth, our sample included men and women aged between 20 and 69 years. Since age and sex could influence the gut microbiome, a more homogenous sample should be targeted.

Nevertheless, the results of the current PROVIT trial are promising, as little is known about gene expression changes after randomized, placebo-controlled probiotics studies. Further holistic, integrative analyses of probiotics RCTs on Multi-Omics-Level (genomics, transcriptomics, metabolomics and proteomics) are necessary to further investigate the mechanisms of the gut-brain-axis as systems biology approaches are currently still lacking. Our current preliminary analysis of the PROVIT sample, which is a precious resource for further secondary analyses as it contains DNA, RNA and 16SRNA sequencing of stool samples before and after the probiotics RCT, gives first insights in gene expression changes of three target genes. Regardless, those gene expression results are very limited as the three target genes have hundreds of protein-protein interactions, which were depicted in the network analysis – see Section 3.3. As shown in Figure 4, IL-6 is interacting with six downstream proteins, namely SH3GL2, FAM20C, ZBTB16, IL6ST, HRH1 and IL6R. SH3GL2, also called Endophilin-A1, has been implicated in synaptic vesicle endocytosis and might recruit other proteins to membranes with high curvature. The Golgi serine/threonine protein kinase (FAM20C) is known to phosphorylate secretory pathway proteins and plays a key role in biomineralization of bones and teeth [73,74]. Moreover, it also plays a role in lipid homeostasis, wound healing and cell migration and adhesion [75]. The ZBTB16 protein (Zinc finger and BTB domain), acts as a transcriptional repressor [76] and may play a role in myeloid maturation and in the development and/or maintenance of other differentiated tissues. Probable substrate-recognition component of an E3 ubiquitin-protein ligase complex which mediates the ubiquitination and subsequent proteasomal degradation of target proteins [77]. Moreover, amongst others, the H1 part of the HRH1 protein mediates neurotransmission in the central nervous system. IL6ST (Interleukin 6 signal transducer) and IL6R (Interleukin 6 receptor) are closely connected with IL-6. The binding of IL-6 to IL6R induces homodimerization of IL6ST and thus the formation of a high-affinity receptor complex. This initializes an activation cascade that mediates signals which regulate immune response, hematopoiesis, pain control and bone metabolism (by similarity).

The differentially expressed target gene IL-6 is concatenated with the other two target genes and hundreds of further genes, which have functions in building up cellular components, organization of biogenesis, developmental processes & growth, immune system processes, locomotion, metabolic processes, rhythmic processes, signaling and single-organism processes. Therefore, probiotics could affect the gene expression of hundreds of targeted genes, which cannot be analyzed with qPCR. To describe the effects of probiotics in a holistic way, we will expand the molecular biological analyses in the follow-up project by more detailed genome-wide transcriptomics and metabolomics analyses. We will analyze the concatenation between gut-bacteria-diversity (measured with 16SRNA sequencing), gut-bacteria metabolites (measured with NMR analyses) and gene expression (measured with RNA-seq or gene expression array technology) of the mentioned PPI network gene groups in the future, follow-up project.

## 5. Conclusions

The current project gives first insights in favorable gene expression changes of IL-6 in fasting blood of depressed patients after a double-blind, placebo-controlled probiotics RCT. As probiotics seem to have positive effects on the mild chronic inflammation processes in MDD, psychobiotics could open a new avenue of treatment in psychiatry. In future, we might use probiotics as an additional treatment in depressed patients to influence the course of disease positively.

## Figures and Tables

**Figure 1 nutrients-12-02575-f001:**
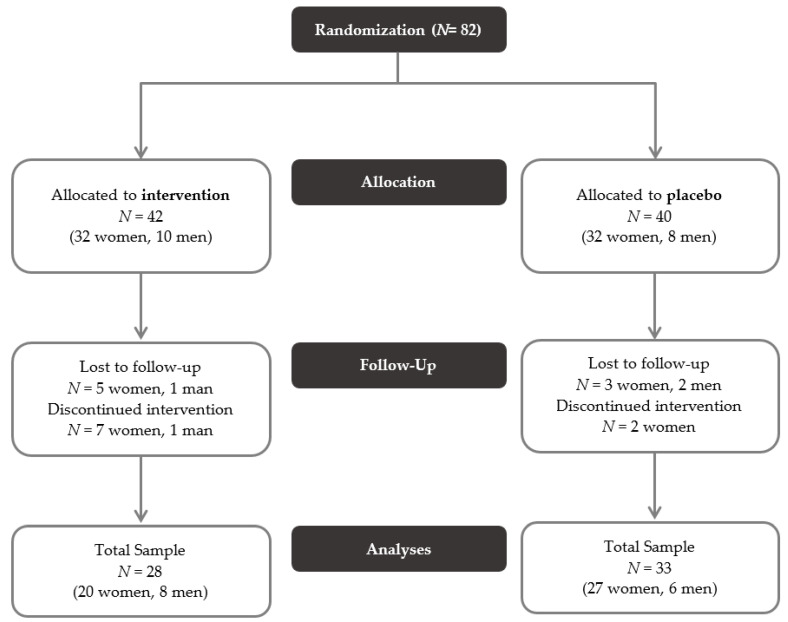
CONSORT flow diagram of the PROVIT study.

**Figure 2 nutrients-12-02575-f002:**
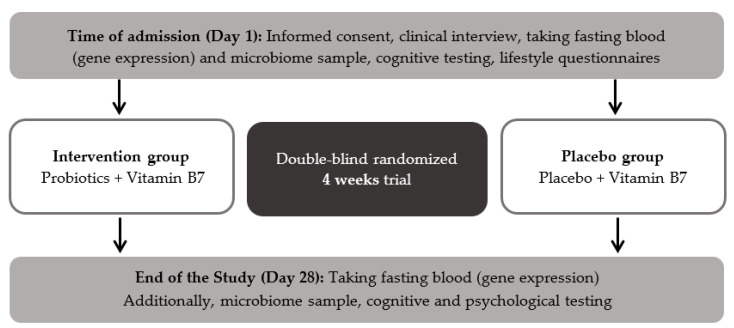
Workflow of the PROVIT study.

**Figure 3 nutrients-12-02575-f003:**
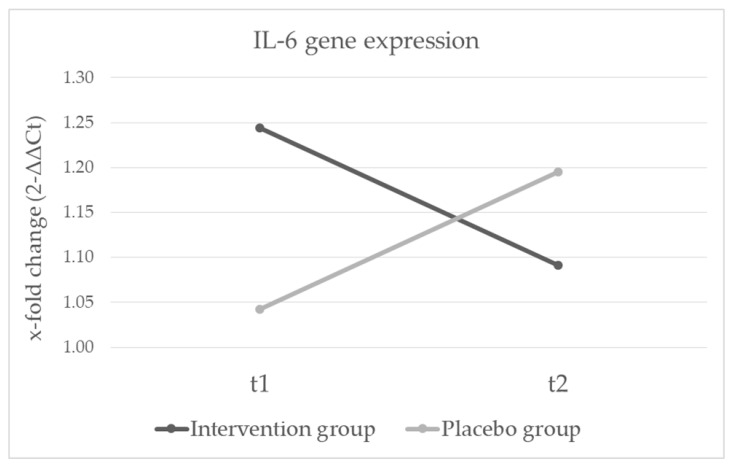
Changes in *IL-6* gene expression after four weeks of probiotic treatment in individuals with depression (time x group interaction *p* = 0.022).

**Figure 4 nutrients-12-02575-f004:**
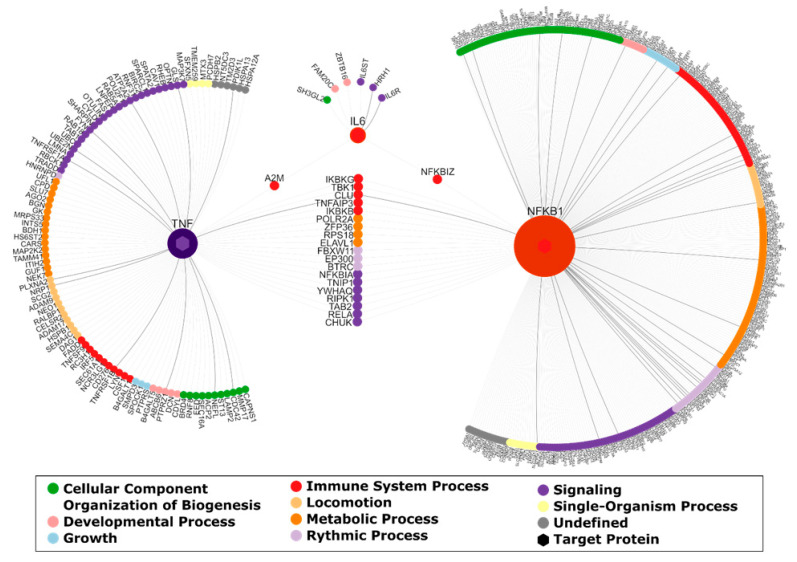
Brain-PPI (protein–protein interaction) network for interleukin-6 (IL-6), tumor necrosis factor (TNF) and nuclear factor kappa B subunit 1 (NFKB1). Interactions between proteins are visualized by edges. Edges that are associated with diseases of mental health are highlighted in black. Moreover, each protein is colored according to their top-level biological process. The degree of the target proteins is emphasized by node size.

**Figure 5 nutrients-12-02575-f005:**
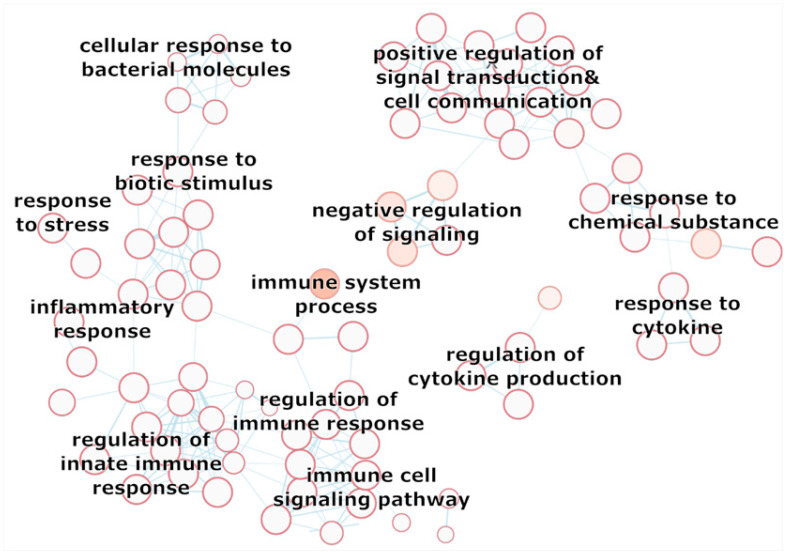
Selection of the Enrichment Map based on the enrichment of GO biological processes of the brain-interaction network of *IL-6, TNF, NFKB1*. The interacting gen set was queried in g:Profiler, resulting in a enrichment of biological processes. The Cytoscape Enrichment Map App was used to generate the corresponding visualization of GO terms (nodes) and their gene set overlap presented as edges. Moreover, each protein is colored according to their top-level biological process. The degree of the target proteins is emphasized by node size.

**Table 1 nutrients-12-02575-t001:** Description of the PROVIT sample at baseline.

	Intervention Group (*N* = 28)	Placebo Group (*N* = 33)	Statistics	
	*N (%)*	*N (%)*	*χ²*	*p*
Sex (female)	20 (71.4 %)	27 (81.8 %)	0.925	0.336
Smoking (yes)	9 (32.1 %)	19 (57.6%)	3.946	0.047
Dairy products before trial (yes)	12 (48.0 %)	10 (34.5 %)	1.016	0.313
	*Mean (SD)*	*Mean (SD)*	*T*	*p*
Age (years)	43.00 (14.31)	40.11 (11.45)	−0.876	0.384
Waist-to-hip ratio	0.86 (0.07)	0.84 (0.10)	−0.739	0.463
	*Median* *(Mean rank)*	*Median* *(Mean rank)*	*U*	*p*
Education (years)	9.00 (31.96)	9.00 (30.18)	435	0.665
Illness duration (years)	6.00 (27.92)	10.00 (31.64)	375	0.409
BMI [kg/m²]	23.96 (32.43)	25.89 (29.79)	422	0.563

Note. SD = Standard Deviation, BMI = Body Mass Index.

**Table 2 nutrients-12-02575-t002:** Medication of PROVIT sample at baseline (in alphabetical order).

	Intervention Group (*N* = 28)		Placebo Group (*N* = 33)	
	*N*	*%*	*N*	*%*
Anticonvulsants	3	10.7	3	9.1
Atypical antipsychotics	10	35.7	9	27.3
Benzodiazepines and hypnotics	5	17.9	7	21.2
Glutamatergic antidepressants	0	0.0	1	3.0
Low potency antipsychotics	1	3.6	7	21.2
Melatonin-like antidepressants	0	0.0	1	3.0
Mixed preparation of antidepressant and antipsychotic	1	3.6	0	0.0
Noradrenergic and specific serotonergic antidepressants	3	10.7	3	9.1
Norepinephrine–dopamine reuptake inhibitor (NDRI)	1	3.6	5	15.2
Selective serotonin reuptake inhibitors (SSRIs)	16	57.1	25	75.8
Serotonin–norepinephrine reuptake inhibitors (SNRIs)	7	35.0	12	36.4
Tri- and tetracyclic antidepressants (TZA)	0	0.0	2	6.1

**Table 3 nutrients-12-02575-t003:** Mean values and standard deviations in depression scores.

	Intervention Group (*N* = 28)		Placebo Group (*N* = 30)	
	*Mean*	*SD*	*Mean*	*SD*
BDI-II t1	30.75	8.40	32.60	10.93
BDI-II t2	15.11	7.91	18.20	11.53
HAMD t1	15.07	6.32	14.43	4.41
HAMD t2	9.11	5.16	8.13	6.16

Note. SD = Standard Deviation, BDI-II = Beck’s Depression Inventory, HAMD = Hamilton Depression Scale.

**Table 4 nutrients-12-02575-t004:** Results in *IL-6* gene expression after four weeks of probiotic treatment.

	Intervention Group (*N* = 21)	Placebo Group (*N* = 25)	Time		Group		Interaction	
	*Mean (SD)*	*Mean (SD)*	*F*	*p*	*F*	*p*	*F*	*p*
IL-6 t1	1.24 (0.49)	1.04 (0.50)	0.000	0.997	0.133	0.717	5.67	0.022
IL-6 t2	1.09 (0.38)	1.20 (0.60)						
TNF t1	1.05 (0.27)	1.04 (0.27)	0.861	0.568	0.001	0.971	0.035	0.853
TNF t2	1.08 (0.37)	1.09 (0.29)						
NFKB1 t1	1.02 (0.16)	1.02 (0.17)	0.331	0.359	0.062	0.804	0.674	0.416
NFKB1 t2	0.99 (0.14)	1.02 (0.21)						

Note. IL-6 = Interleukin-6, TNF = tumor necrosis factor, NFKB1 = nuclear factor kappa B subunit 1, SD = Standard Deviation.

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
