# Peer review of "Interleukin-6 Gene Expression Changes after a 4-Week Intake of a Multispecies Probiotic in Major Depressive Disorder—Preliminary Results of the PROVIT Study"

_nutrients, 2020, doi:10.3390/nu12092575_

Round 1

Reviewer 1 Report

This is an interesting well-written paper which is of value to the field. The major comment that needs attending to is regarding the 'data elephant in the room'. 

The focus of the paper is the cytokine gene expression changes, which is fine, however where is the analysis or mention of this interfacing with change on depression scores over time??

This would be the major finding of this work. Whether this is analysed via a regression or as the primary outcome with the cytokine gene expression as covariates or mediating factors. If being published elsewhere at least comment on this.

Other minor comments:

1) Can't really say that the probiotics are a useful treatment for MDD unless this depression outcome data is presented (regardless of showing the potential link re anti-inflammatory effects). If not presenting this data then temper conclusion

2) If mentioning the Akkasheh paper in the intro it is worthwhile contrasting this with the Rucklidge MDD paper

3) Was there a minimum HAMD level for inclusion? Usually there is for MDD studies. Also, the HAMD and BDI levels seem to be quite different

4) Why was B7 used in the placebo? Seems odd and not sure why due to 'ethical reasons'

5) Some discussion on what the individual strains are in a mental health context would be good. 

6) Just clarifying re null Time effect- would this not be normally done per group individually? Looks like just one analysis for both

7) For Figure three it would be good to detail P value in the footnote or chart

8) Seeing only 6 downstream proteins affected by IL-6 according to Figure 4- would be good for brief discussion on what they are

9) Also, a bit more detail on how your Probiotic combination may be different or superior to that used by other studies would be beneficial

Reviewer 2 Report

The manuscript is well written and the data, of high quality, convincing and well presented, justify its conclusions.

However, there are minor typos or curiosities, that should be revised, explained and, if necessary, corrected.

Minor points:
1. On line 83 and line 95 it´s repeated, exactly, the same paragraph. Is this correct?
2. On line 111, it´s written TNF-a. Is that correct?
3. On line 124, which are the strains?
4. On line 128, the authors quote a study, PROVIT, which has not yet been accepted for publication (Reference 52 [in submission]). As this publication, and therefore its results, are not yet accepted I would advise to remove it from the bibliography.
5. The acronym MINI is written for the first time on line 140 but the meaning is explained on line 160. Please, move this explanation to line 140.
6. Figure 1 is based on the CONSORT guide, but the reference is missing in the bibliography. Please, introduce the corresponding reference in the bibliography.
7. Please, clarify which is the concentration of the probiotic strains. It is not clear if the concentration is at least 2.5 x 109 CFU/g of each strain or at least 7.5 CFU /3 g for the mixture.
8. Has the manufacturer characterized the different strains?
9. In addition to these strains, are other bacterial strains included? line 181
10. Does the Omni-biotic strees repair® require refrigeration?
11. Did the intake of this product during the study cause any adverse effect?
12. On line 298, “Previous studies investigating other clinicals samples showed that probiotics could affect IL6 levels…..”.Which are those studies? Neither Borzabadi et al. (2018) nor Tabrizi et al. (2019) observed a decrease in IL6 levels.
13. On line 312, which is the meaning of the acronym CPR?
CONCLUSION
I suppose we will have to wait for the full PROVIT study before we can draw conclusions about the use of probiotics as a treatment for psychiatric disorders.

Reviewer 3 Report

 My concern is the statistics. The authors indicated in the abstract  main interaction and main effects, it is unclear which groups or factors? It is not indicated in the statistical section as well. It looks like two-way ANOVA, if so what are the two factors.

The authors need to clearly demonstrate in the statistical section what statistical strategy used to compare control and intervention groups for gene expression.

In the statistical section, differences between two groups were analysed with t- test,Mann-Whitney-U test or Chi-Square test based on normality distribution. Please indicate clearly which parameters were used for which test, and this contradicts with the stats reported in the abstract which seems to me like a two-way ANOVA. Please also indicate statistical analysis used for microbiome and network analysis.

In my opinion, it is essential the authors update the statistical section in detail in order for me to review the remaining manuscript as the stats are crucial for data intrepretation. My major concern is the stats for gene expression. In the resuls section, no indication of whether they used two-way ANOVA but reports seems like two-way ANOVA, please confirm. For two-way ANOVA, while the interaction is significant, the main effects are also non-significant, and so statistically, authors cannot clain IL-6 expression level is decreased as this is misleading.

I look forward for a revised statistical section .

Round 2

Reviewer 3 Report

Thank you for your revised version.

A few more questions.

1) Can you please do simple main effects (post-hoc) between the groups at t1 and t2 and report the mean differences with p-value.

Also, can you reword this sentence under statistics as it is unclear.

"Changes between baseline and end of the study were calculated with repeated measures two-way analysis of variances (ANOVA) independent factors time point and group, and dependent variable scores in BDI-II and HAMD respectively levels of gene expression in IL-6, TNF and NFKB1.

Also include in the methods, post-hoc analysis (indicate which post-hoc used, preferably Tukey).

2) What are the housekeeper(s) used for PCR? If you used more than one, did you calculate the gene expression based om geomean of the housekeepers. Importantly, how did you determine/choose the housekeepers is the most suitable?

3) It is very uncommon to self-cite a manuscript under revision (53). Please remove this.

4) Lastly, I strongly would not recommend to say in the last sentence  of your conclusion" ....additional treatment in depressed patients to lower IL-6 levels..."  The statistical outcome is not strong to support this claim. While there is a significant interaction, and if the main simple analysis do not provide a significant difference, I would not claim this. It also depends on the housekeepers used, single or multiple. So, please re-phrase.

5)I suggest the authors to amend their title as it needs to align with the statistical outcome.

Author Response

Dear Reviewer,
